# Adaptive Agricultural Strategies for Facing Water Deficit in Sweet Maize Production: A Case Study of a Semi-Arid Mediterranean Region



Lea Piscitelli [1,*] , Milica Colovic [1,2], Adel Aly [1] , Mohamad Hamze [1,3] , Mladen Todorovic [1] , Vito Cantore [4] and Rossella Albrizio [5]

1 CIHEAM-Bari, Via Ceglie 9, 70010 Valenzano, Italy; m.colovic94@gmail.com (M.C.); aly@iamb.it (A.A.); mohamad.hamze@inrae.fr (M.H.); mladen@iamb.it (M.T.)
2 Department of Soil, Plant and Food Sciences, University of Bari Aldo Moro, Via G. Amendola 165/a, 70126 Bari, Italy
3 CIRAD, CNRS, INRAE, TETIS, University of Montpellier, AgroParisTech, CEDEX 5, 34093 Montpellier, France
4 Institute of Sciences of Food Production, National Research Council (CNR-ISPA), Via Amendola, 122/O, 70125 Bari, Italy; vito.cantore@ispa.cnr.it
5 Institute for Agricultural and Forestry Systems in the Mediterranean, National Research Council (CNR-ISAFOM), Piazzale Enrico Fermi 1, 80055 Portici, Italy; rossella.albrizio@cnr.it
* Correspondence: piscitelli@iamb.it

**Abstract:** Maize is a crucial global commodity, which is used not only for food, but also as an alternative crop in biogas production and as a major energy-supply ingredient in animal diets. However, climate change is jeopardizing current maize production due to its direct impact on weather instability and water availability or its indirect effects on regional climate suitability loss. Hence, new areas for sweet maize cultivation should be considered in the future. Therefore, this study focuses on the possibility of producing maize in a challenging environment in Southern Italy considering rainfed cultivation and two irrigation regimes (full and deficit). The experiment was conducted during two subsequent growing seasons under semi-arid Mediterranean climate conditions. The overall results indicated a significant difference in biomass and yield between irrigated and non-irrigated treatments, and between full and deficit irrigation. Sweet maize cultivated under deficit irrigation gained less biomass than under full irrigation and its development and fruit maturation were delayed. Under deficit irrigation, the plants gave lower yields and a higher percentage of the panicle weight consisted of kernels. Irrigation water productivity was higher for deficit than for full irrigated treatment. These findings indicate the feasibility of sweet maize production in semi-arid areas of Southern Italy using adaptive agricultural strategies including deficit irrigation and controlled water stress. Given the importance of maize production, understanding of maize growth and productivity in a challenging environment may support future agricultural programming and thereby contribute e to mitigation of the direct and indirect effects of climate change.

**Keywords:** climate change; deficit irrigation; rainfed cultivation; maize development; irrigation water productivity; Southern Italy

## 1. Introduction

Maize (*Zea mays* L.) is a worldwide important staple food—a high-yield commodity crop that can be produced in a wide range of environmental conditions [1]. Nowadays, 170 countries in the world produce maize and its production is steadily increasing. More than 239 million hectares are cultivated with maize and about 1.4 billion tons have been produced only in 2019 [2]. Besides for food consumption, maize is used as an alternative crop in biogas production and as a major energy-supply ingredient in animal diets [3].

Lobell et al. [4] examined more than 20,000 maize trials over 8 years and reported that for each degree of air temperature increase above 30 °C, maize yield was reduced by

approximately 1.4% compared to the optimal conditions for growing in terms of both water-supply and temperature range. Thus, the yield was reduced due to shortening of growing season, especially during the yield formation stage, and the evapotranspiration decreased as it could not be compensated by adequate water supply [4]. Moreover, sweet maize yield is drastically affected by intra-seasonal weather variability and soil moisture conditions during the cropping season [5]. According to the World Meteorological Organization, the 2019 closed a decade of extraordinary high air temperatures at the global level; if not stopped this trend could lead to the temperature rise of 3 to 5 °C by the year 2100. Moreover, in the future, a reduction of water availability for agriculture is expected [6] with a direct impact on maize production and indirect alteration of regional climate-related suitability [7].

The dendroclimatology has already recorded historical changes in some agricultural areas due to crop vulnerability to climate variation, and certainly the case of grain in North Africa during the Roman Empire is an effective example [8]. It is a matter of fact that Romans did not stop producing grain but shifted the production to more suitable regions and adopted adaptive agricultural strategies. Nowadays, the choice of genetic breeding capable to increase crop resistance to water/heat stress together with the adoption of other adaptive agricultural practices are valuable opportunities to tackle the direct and indirect impacts of climate change on agriculture. Some maize hybrids show a buffering capacity against drought stress resulting in a minimized yield reduction [9–11]. Moreover, the acceptance of other adaptive agricultural strategies as water saving cropping methods, shifting of the planting date, increasing the soil coverage have demonstrated to build up the tolerance of maize against water deficit [12].

In this sense, Italy can be a quite interesting case study because of the extreme climatic differences between Northern and Southern regions and high water requirements of sweet maize [13]. According to [14], the Northern Italy regions have the best characteristics for maize production, and this is corroborated by Istituto Nazionale di Statistica (ISTAT) data. Indeed, in 2019 and 2020, the Northern regions accounted for more than 88% of the total cultivated surface and more than 90% of the total maize national production. Nevertheless, a minimum maize production is present in Southern regions and irrigation management could be a powerful adaptive strategy among others. Despite the high costs to be incurred for irrigating maize in water-scarce areas, including Southern Italian regions, this practice or its use among other management practices may maintain or increase the crop production and cope with the challenging climatic conditions [15]. Therefore, adopting the adaptive strategies for contemporary agriculture is necessary to cope with extreme weather events. In this context, the testing of different options to cultivate maize in challenging environmental condition may contribute to oppose the global production reduction due to decrease of the areas suitable for production.

So far, numerous modeling studies based on real data and projections have been carried out to evaluate the adaptation of maize cultivation to climate change all over the world [16–20]. However, only few studies focused on the sweet maize cultivation [21–23]. Therefore, in this study, the agricultural performance of sweet maize has been tested during two cropping seasons under semi-arid climatic condition of Southern Italy. Three different water regimes were applied aiming to understand the most suitable adaptive strategy in a Mediterranean environment characterized by high variability of precipitation and frequent shortage of irrigation water.

## 2. Materials and Methods

### 2.1. Experimental Site

The experiment took place in Valenzano (Bari) (41°03′16″ N 16°52′33″ E), Apulia Region (Southern Italy) under Mediterranean climate, characterized by 30-year average annual precipitation of about 550 mm, and maximum air temperature reaching 30–35 °C in summer. During the two growing seasons (2019 and 2020), the weather data were recorded

by the sensors of an agro-meteorological station placed in a surface of 400 m$^2$ covered by perennial grass and located close to the experimental field.

Soil samples were taken before the beginning of the experiment at 10 to 35 cm depth. Collected soil samples have been air dried and sieved at 2 mm before chemical characterization. Soil is classified as silty-clay-loam with high carbonates content and poor in organic carbon (Table 1).

**Table 1.** Physical and chemical characteristics of the soil.

| | | |
|---|---|---|
| Stones and gravel | | 75 |
| Sand | g/kg | 170 |
| Clay | | 234 |
| Silt | | 596 |
| Textural Class (USDA) | | Silty loam |
| pH (H$_2$O); 1:2.5 | | 8.1 |
| pH (CaCl$_2$); 1:2.5 | | 7.6 |
| Electrical conductivity 1:2 at 25 °C | dS/m | 0.24 |
| Total Carbonate | | 55 |
| Organic C | g/kg | 11.6 |
| Total N | | 0.9 |
| Available P | mg/kg | 17 |
| K$^+$ exchangeable | | 465 |

Sweet maize (*Zea mays* var. *saccharata* L., hybrid Centurion F1) was cultivated during two subsequent growing seasons, with a density of 10 plants per m$^2$ (0.5 m distance between rows and 0.2 m on the row). Sweet maize was sown on 2nd and 3rd May in 2019, while in 2020, because of the restriction imposed by national law for contrasting SARS-CoV-2 pandemic, its crop cycle was postponed, and the transplanting was preferred to sowing. Seeds were sown in growing trays in the nursery on 18th of May 2020, and then the seedlings were moved to the field on 16th of June.

### 2.2. Experimental Design

Within a more complex split-plot experimental design carried out to analyze maize response to different water and nitrogen supply, in this study only nine plots of 10 × 10 m (100 m$^2$) were investigated. These 9 plots had three irrigation scheduling treatments with 3 replicates each: (i) full irrigation (FI); (ii) deficit irrigation (DI), applying 50% of full irrigation requirements; (iii) rainfed cultivation (RC). Crop water requirements and irrigation scheduling were managed on a daily basis using an Excel-based irrigation tool [24] following the standard FAO Penman-Monteith approach [25]. In full and deficit irrigation plots, water was supplied through a drip irrigation system. Irrigation was supplied 8 and 12 times in 2019 and 2020, respectively, with the corresponding irrigation volumes during the crop growing cycle of 2811 and 2912 m$^3$ ha$^{-1}$ in FI treatment, while half of these volumes were applied in DI treatment.

### 2.3. Agronomic Parameters

At about 90 days from sowing, secondary roots, stalk tillers, foliage cover, number of internodes and tassel ramification were measured and reported according to the International Board for Plant Genetic Resources [26] with the aim to monitor the possible occurrence of stress symptoms. The phenological stages were recorded and reported using the code system proposed by [27] at about 100 days from sowing.

At the stage of panicle maturity, a total of 10 plants per plot were sampled. Stems plus leaves and roots were divided and dried in the oven and the weights were recorded.

Panicles were divided into marketable and non-marketable, and their fresh weights were recorded. Only for marketable panicles kernel and cob were separated and dried in the oven. Irrigation water productivity (IWP, kg m$^{-3}$) was determined for FI and DI and

expressed as the ratio between the total dry vegetative biomass, or fresh yields (panicle and kernel) and the total supplied water (irrigation volume).

### 2.4. Statistical Analysis

Descriptive statistic was used for explaining the frequency of some categorical or ordinal descriptors. Analysis of variance was conducted among treatments in the same year, and significance of differences among treatments were separated using Fisher's Least Significant Difference (LSD) at a 5% probability level. In graphs and tables, the means with significant differences ($p \leq 0.05$) are labelled with different letters while the values with no significant differences are reported with no labelling letters.

### 3. Results and Discussion

In Figure 1 are presented the values of monthly average temperature and precipitation recorded during the crop growing cycles in 2019 and 2020. The highest daily average temperature was recorded in the first ten days of July in 2019 (30.0 °C) and at the beginning of August in 2020 (28.9 °C), while the lowest daily average temperature was detected in the ten first days of May in 2019 (11.6 °C) and in the middle June in 2020 (19.5 °C).

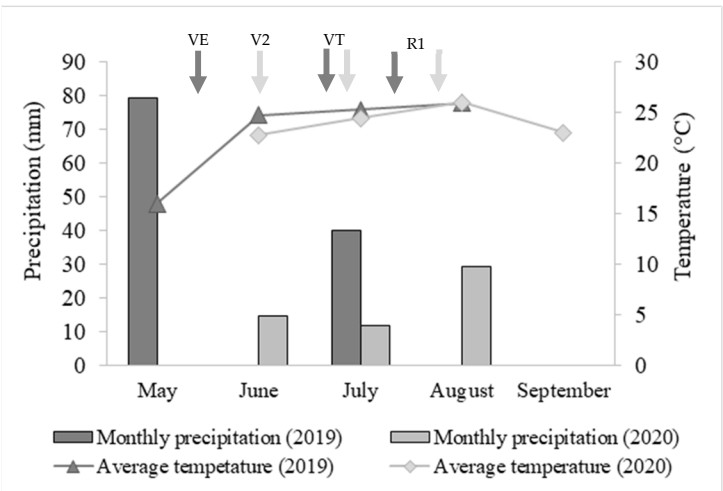

**Figure 1.** Monthly variation of precipitation and average air temperature during the growing seasons in 2019 and 2020. Dark arrows indicate the phenological stages for 2019 and light grey arrows for 2020. The codes indicate: VE: emergence, V2: second leaf visible, VT: tasseling stage, R1: silking.

In the 2019 growing season, the total precipitation was 119.3 mm, mainly concentrated in May, and two moderate rainy events in the first half of July. The total precipitation in 2020 was about the half of 2019 (55.8 mm), distributed in June, July, and August with a unique heavy rain event at the beginning of August. The irregularity of rain events and the occurrence of high temperature confirm the typical Mediterranean conditions where the impact on maize yield and quality is relevant due to the severe stress caused by the concurrent effects of drought and heat [28].

The length of growing cycle was similar in two growing seasons despite the differences in air temperature and precipitation. The crop growing cycle lasted 102 days in 2019 and 105 days in 2020 (15 days in greenhouse and 90 days in the field). Although the difference in weather conditions in two years does not affect the duration of crop cycle, the impact on maize growth under different water regimes was important. Indeed, in both growing seasons, the irrigated plants reached the reproductive stage of maturity, and no decline of growth was recorded regardless the volume of water supplied. Differently, the maize growth under RC was reduced during both crop growing cycles and, at harvesting, the plants was small and malformed.

In Figure 2 is reported the percentage of plants ascribable to the phenological stage detected at 100 days after sowing/transplanting and identified according to the classification proposed by [27] for maize growth and development.

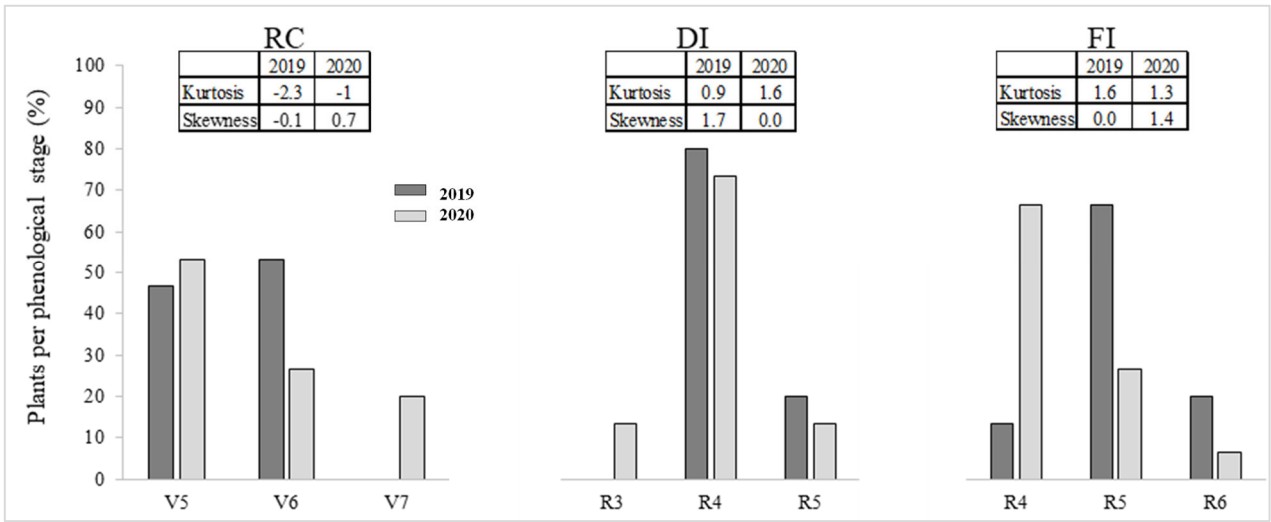

**Figure 2.** Percentage of plants ascribed to different phenological stage under three water treatments at 100 days from sowing. Dark grey column stands for 2019 cropping season while light grey for 2020. The classification refers to the codification system proposed by [25]. Specifically, V indicates the vegetative phenological stage of the plants while the numbers show the number of well-developed leaves; R indicates reproductive stage, and the numbers refer to the consistency of kernels (3 = milk; 4 = dough; 5 = dent; 6 = maturity).

In the case of FI, the percentage of plants reaching the three phenological stages was different in two cropping seasons. In 2019, most of the panicles reached the dough stage of consistency (R4), while in 2020 only 27% of plants reached the dough stage and 67% finished the milk stage (R3). The frequency distributions in both yeas were leptokurtic (kurtosis > 0). In 2019, the distribution showed a perfect normal distribution (skewness = 0), while in 2020 a lognormal distribution was observed (skewness > 0). For DI treatment, the frequency distribution in 2019 and 2020 do not differ (both leptokurtic) and most of the plants at the harvesting time had panicles in the late milk stage. In opposition to FI, the skewness in DI indicated a normal distribution in 2020 and a lognormal distribution in 2019. A completely different situation was observed in the case of rainfed cultivation where the plants of about 100-days old failed to reach the reproductive stage (V) and the frequency distribution in both seasons was platykurtic (kurtosis < 0): positively asymmetric in 2019 and negatively in 2020. Water deficit was recognized as a key factor for slowing-down maize growth and development [29]. This is particularly visible under rainfed cultivation but determines the slower kernel maturation of DI compared with FI.

The differences between irrigated and rainfed treatments were properly highlighted through the observations of plants in the field and the registration of some crop categorical descriptors (Table 2). Based on these results, the plants under FI treatment showed a full development with large foliar cover, the presence of stalk tillers and formation of secondary roots during both growing seasons. The plants under DI treatment had a reduced foliage cover compared with FI and produced stalk tillers only during the 2020. On the other hand, the RC plants had a scarce development.

**Table 2.** Morphological traits measured at harvesting. Values are expressed as percentage over the total plants per plot.

| Year | Treatments | Secondary Roots (%) | Stalk Tillers (%) | Foliage Cover |
|---|---|---|---|---|
| 2019 | FI | 100 | 100 | Large |
|  | DI | 87 | 60 | intermediate |
|  | RC | 0 | 0 | Small |
| 2020 | FI | 27 | 33 | Large |
|  | DI | 47 | 0 | intermediate |
|  | RC | 0 | 0 | Small |

Secondary roots favor plant anchorage for preventing lodging and may contribute to nutrient and water uptake [30]. However, in this study no lodging has been recorded in treatments. Secondary roots development is typical of adult-to-reproductive morphological transition [31] and, in this sense, the results provided in Table 2 agree with those presented in Figure 2. Specifically, even in this case, the RC plants demonstrated 100% reduction of growth, while DI and FI reached the reproductive maturities with well-developed panicles.

Stalk tillers formation is controlled by a complex gene regulatory system but conventionally their development is a signal of favorable growing condition [32]. Indeed, the FI treatment corresponding to the optimal soil water conditions reported the highest stalk tillers number in 2019 and in 2020. However, in maize, tillers may occur in late stages and, therefore, do not participate to the overall productivity. According to [33], tillers emergency may lead to soil water depletion and resulted in less available water for the kernel formation. In fact, their formation is a genetic heritage of Teosinte domestication into Maize [34] and they are not considered to affect stalk development when water or nutrients are not limiting [35,36]. However, it can be a major concern in dryland condition or under low plant density [35,37]. Moreover, according to [38], the farmers use to remove these tillers to overcome the possible negative effects on the yield in the latter field conditions.

Maize capacity to intercept light depends on several factors, including plants spatial arrangement, canopy architecture, foliage structure, leaves number and photosynthetic potential [39]. In turns, light interception is an important factor in yield determination [40]. Similarly, to the previous parameters, the foliage cover demonstrated higher vigor of fully irrigated plants than for other two treatments (Table 2). Although, the large foliage cover can act directly for the preservation of soil moisture [41], Fang et al. [42] assert that reducing plant density may lead to decrease of crop water use.

According to [43], maize plants have generally 13 leaves, one per internode. In their study, the plants reached 11 leaves under irrigated condition and 10 under water deficit stress. Table 3 shows that plants of maize reached at most 11 leaves indifferently from the water regimes and the growing season. In 2019, most of the plants reached the 11th leaves in full and deficit irrigated treatments. During 2020, in FI and DI, only 40% of plants had the same performances, while the others were homogeneously distributed among 8, 9 and 10 values. It confirmed the most stressful conditions encountered by maize in the 2020 season, as the crop had to cope with rainfall scarcity during the vegetative stage. Moreover, it is well known that leaves formation and expansion is greatly affected by water deficit [44,45], as it reduces the internode extension [46] and, more generally, it is an inhibitory condition for maize development [47].

Plants with a maximum number of internodes lower than the potential is one of consequences that has to be accepted when cultivating maize far from its optimal pedoclimatic conditions.

Tassel is defined as male inflorescence and can present a certain level of ramification. Maize plants in DI had mostly a primary ramification (80% in 2019 and >90% in 2020), while in FI during 2019 all the plants developed a tassel of tertiary ramification level and in 2020 the repartition was more variable (primary ramification 40%, secondary ramification 47%, and tertiary 10%). Proper tassel development is essential for food production [48], but

smaller tassel reveals positive correlation with higher production probably due to better transition of light [49].

**Table 3.** Percentage of plants recorded to have a certain number of internodes.

| | RC | | | | DI | | | | FI | | |
|---|---|---|---|---|---|---|---|---|---|---|---|
| N° of Internodes | Percentages of Plants | | N° of Internodes | Percentages of Plants | | N° of Internodes | Percentages of Plants | |
| | 2019 | 2020 | | 2019 | 2020 | | 2019 | 2020 |
| 5 | 47 | 53 | 8 | 0 | 20 | 8 | 0 | 20 |
| 6 | 53 | 27 | 9 | 7 | 13 | 9 | 0 | 20 |
| 7 | 0 | 20 | 10 | 13 | 27 | 10 | 7 | 20 |
| | | | 11 | 80 | 40 | 11 | 93 | 40 |

Figure 3 shows the total dry biomass of maize consisting in roots and stems plus leaves for the three water treatments during both growing seasons. Total dry biomass of RC treatment was 17 and 18 g plant$^{-1}$ in 2019 and 2020, respectively, i.e., significantly lower than the values recorded in irrigated treatments. Dry stems plus leaves values of irrigated treatments were lower in DI than in FI, but not significantly different from each other in 2020 and significantly different in 2019. It is well documented that water deficit may reduce maize dry matter accumulation, internode extension, slow down dry matter accumulation, and alter the morphology [46,50–52]. Considering root weights, the different weather conditions during two growing seasons affected their value more than water inputs. According to [53], the maize plants react to water deficit by translocating resources (water and nutrients) from stems to roots. In turn, translocation caused an enhanced roots growth and inhibited stems biomass accumulation [54].

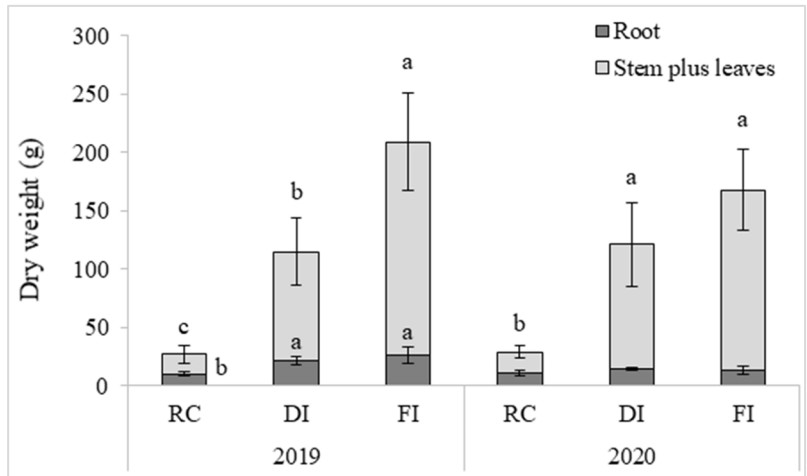

**Figure 3.** Maize plant stems plus leaves (light grey) and roots dry weight (dark grey). Treatments were statistically analyzed, and data with different letters indicate significant differences at ($p \leq 0.05$). Vertical bars indicate SD.

As highlighted previously, the development of plants that undergo rainfed condition stopped at vegetative stage, during leaf development and neither elongated the stems. On the opposite, plants in irrigated treatments reached the reproductive stage. Hence, the panicles marketable and not-marketable yields are reported in Table 4.

**Table 4.** Values ($\pm$ SD) of panicle marketable and not marketable yield in full and deficit irrigated treatments. Data with different letters indicates significant difference at ($p \leq 0.05$).

| Year | Treatments | Marketable Yield (t ha$^{-1}$) | Not-Marketable Yield (t ha$^{-1}$) |
|---|---|---|---|
| 2019 | FI | 21.5 $\pm$ 2.8 a | 4.3 $\pm$ 2.0 a |
| | DI | 8.8 $\pm$ 3.2 b | 0.0 $\pm$ 0.0 b |
| 2020 | FI | 22.1 $\pm$ 2.7 a | 3.4 $\pm$ 1.7 |
| | DI | 9.5 $\pm$ 3.5 b | 1.8 $\pm$ 0.9 |

The FI treatment accounted the highest yield in both growing seasons and the yield in DI was less than the half of FI yield. The small difference of FI- yield in two growing seasons can be explained by the intra-seasonal weather variability and the subsequent soil moisture conditions [5]. The high yield gap between DI and FI complies with the findings reported by [55] during dry or normal seasons and it is adequately justified by the high water requirements of sweet maize [13]. Nevertheless, the yield values of DI (10 t ha$^{-1}$) are comparable with those documented by [56] under deficit irrigation of maize cultivated in Central Italy, where the climatic conditions do not differ significantly from that of this experiment. Moreover, the DI yield does not differ from the findings of [57] and [58] in semi-arid region of Zimbabwe and China after the adoption of other different adaptive agricultural practices.

Despite the huge differences between DI and FI in both growing seasons, the almost satisfactory yield reached in DI can be explained by (i) the occurrence of rain events in imminence of flowering stages (VT) in 2020, similarly to the results reported by [59], (ii) high plant density [60], (iii) or drought tolerance response [61]. On the other hand, about 15% of the total yield was identified as not marketable with the only exception of DI treatment in 2019.

In Figure 4, the incidence of kernels to cob weights are reported. Differently from the marketable yield results, the kernel production is higher in DI treatment than in FI in each growing season. According to [62], water deficit decreases cob weight of about 38%, and in parallel it affects kernels weight. On the other hand, Figure 4 illustrates a higher kernel weight under DI as compared to FI treatment during both growing seasons. This result suggests that despite water deficit interferes on maize grain-filling [63], in parallel, it affects kernels to cob distribution shifting the weight gain in favor of kernels.

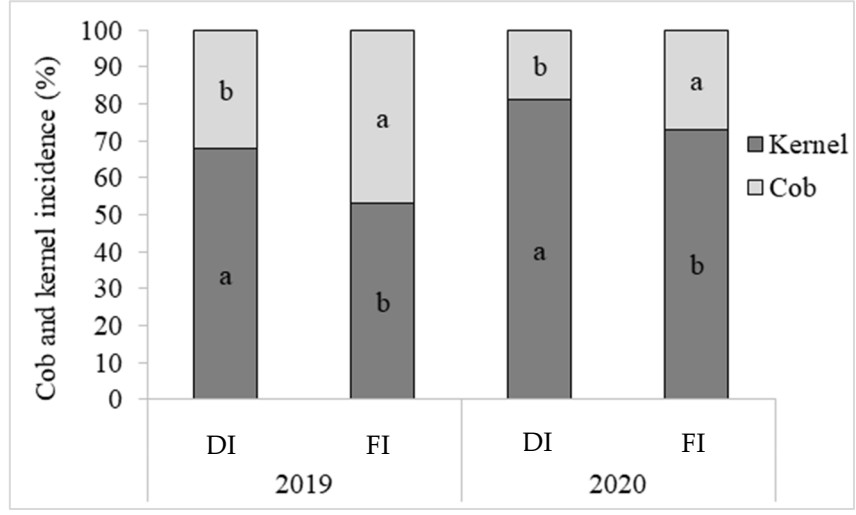

**Figure 4.** Dry kernel and cob incidence (%). Kernels and cobs percentages were statistically analyzed separately. Data with different letters indicate significant difference at ($p \leq 0.05$).

Irrigation water productivity calculated for total dry biomass, fresh panicles and kernels are reported in Figure 5.

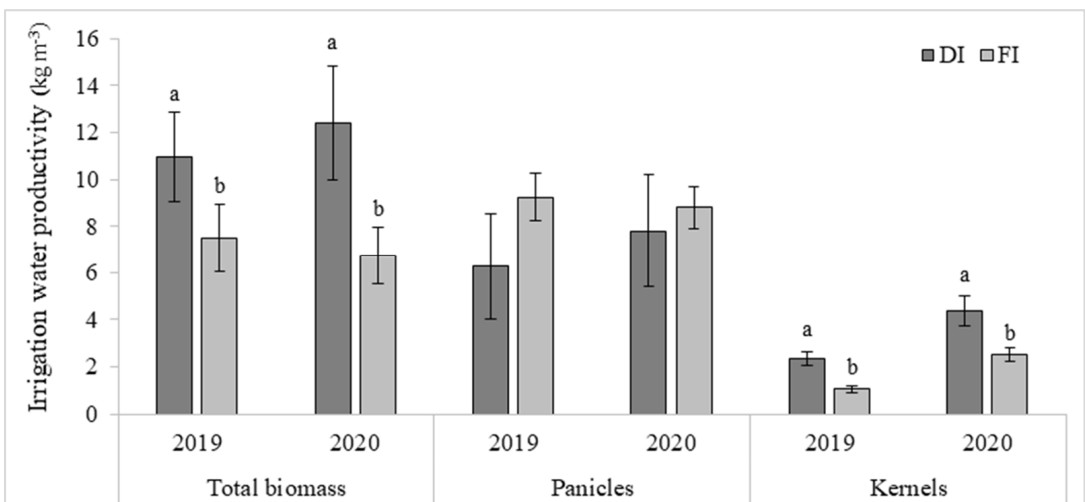

**Figure 5.** Irrigation water productivity (kg m$^{-3}$). Vertical bars indicate standard deviation (SD). Data with different letters indicate significant differences at ($p \leq 0.05$).

Irrigation water productivity was significantly higher in DI treatment than in FI indifferently from the growing season in the case of total biomass and kernels. No significant difference of IWP was observed in the case of panicles. This indicates that maize plants used in this study respond well to moderate water deficit and efficiently use water provided by irrigation, especially in the case of plant biomass and kernels production. During water scarcity, new maize genotypes are capable to balance adequately water use between grain production as plant hydration maintenance [64]. Moreover, [65] observed that even if water shortage is commonly responsible for great maize yield loss, the timing of water deficit can positively affect IWP.

## 4. Conclusions

In this study, the possibility of cultivating maize in new and challenging areas was evaluated. The first finding is that maize cannot be cultivated under rainfed conditions in Southern Italy and other similar climatic areas of the Mediterranean. Moreover, the results indicated that plants cultivated under deficit irrigation gained less biomass and had a predictable slowdown of growth and fruit's maturation. A significantly lower yield was observed under deficit irrigation treatments than under full irrigation. Furthermore, in the case of deficit irrigation, the panicles had higher percentage of the weight consisted of kernels. Understanding the development path and maize productivity in challenging environments and under different water regimes may support future agricultural programing and guide the decision makers in preferring between cultivation for food, fodder, or energetic purposes.

Historically, Southern Italy, as other semi-arid Mediterranean regions, do not represent the areas devoted to the production of maize due to specific environmental conditions and limited precipitation. This was clearly confirmed by yield recorded in this study. However, in the context of changing climate, with extreme weather events and agricultural regions suitability loss, the semi-arid Mediterranean regions could become a maize-producing areas. The use of hybrid and the adoption of few adaptive agricultural strategies have demonstrated that maize can be produced in these regions. Hence, the findings of this study can be extended to all other regions with a typical Mediterranean climate characterized by hot, dry summer and water scarcity.

This experiment considered only the agricultural feasibility of maize production in challenging environment, but an economic evaluation can provide a more wide and

complete standpoint. Additionally, other cultivars and adaptive agricultural strategies have to be arranged and tested to obtain simultaneously the economic profitability and environmental sustainability. Therefore, future studies have to focus on evaluating the arrangement of other adaptive agricultural scenarios as change of planting density, conservation tillage and anticipation of the traditional planting date with the aim to optimize the use of resources and to increase the productivity of agricultural land.

**Author Contributions:** Conceptualization, L.P., R.A., M.T., V.C.; Formal analysis, L.P., M.C., M.H., A.A.; Investigation, L.P., M.C.; Methodology, L.P., A.A.; Visualization, L.P.; Writing—Original draft preparation, L.P., R.A.; Writing—Review and editing, R.A., M.T., V.C. All authors have read and agreed to the published version of the manuscript.

**Funding:** The research was supported by the Master of Science Program in Land and Water Resources Management of CIHEAM Bari (Italy).

**Institutional Review Board Statement:** Not applicable.

**Informed Consent Statement:** Not applicable.

**Data Availability Statement:** The data presented in this study are contained within the article.

**Acknowledgments:** The authors thank Carlo Ranieri (CIHEAM Bari) for technical support during the acquisition of data and Mimmo Tribuzio (CIHEAM Bari) for agronomic assistance in the field.

**Conflicts of Interest:** The authors declare no conflict of interest.

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
