# Peer review of "Adaptive Agricultural Strategies for Facing Water Deficit in Sweet Maize Production: A Case Study of a Semi-Arid Mediterranean Region"

_water, doi:10.3390/w13223285_

Round 1

Reviewer 1 Report

The experiment only has data of two growing seasons. This is a problem. Having data from only two agricultural seasons is, in my opinion, the weakest point of the research.

Another weak point: water and maize prices. This research checks out a water adaptive strategy for maize production. OK. But, what kind of strategy could ignore issues related to efficiency?

Maize crops are one of many alternative commodities for the uses cited by the authors in order to satisfy world demand. There are dozens of alternatives for forage crops or energy crops around the planet, such as barley, soybeans, sorghum, etc. At least, these options could be mentioned by the authors.

We must not forget that maize planting is a very intensive form of crop farming and involves the high use of water.

Apart from the details of the modest covered data of two growing seasons and crops alternatives, the design of the research is correct. The methodology is well developed and explained. There is no objection.

What I saw as missing from this research is a Discussion section. My focus should be on the lack of data due to the short time of collecting results, and the alternatives in order to satisfy the same demand.

At least, I should expect some reference to other alternative crops, water irrigation needs, and crop prices. In order to complete the research. It is a matter to compare efficiency to effectiveness.

Author Response

The experiment only has data of two growing seasons. This is a problem. Having data from only two agricultural seasons is, in my opinion, the weakest point of the research.

Although we could understand the point of view of the reviewer, we should also notice that many agronomic scientific papers largely use a couple of experimental years. Of course, the results would strengthen if we would have more years.

Another weak point: water and maize prices. This research checks out a water adaptive strategy for maize production. OK. But, what kind of strategy could ignore issues related to efficiency?

Thank you for this remark. We did not perform an economic analysis by ourselves, but we strongly believe in the mandatory of this matter, therefore, we included connections to the topic in Introduction (lines 186-189 pag 2) and Conclusions (lines 1080-1084 pag 10).

Maize crops are one of many alternative commodities for the uses cited by the authors in order to satisfy world demand. There are dozens of alternatives for forage crops or energy crops around the planet, such as barley, soybeans, sorghum, etc. At least, these options could be mentioned by the authors.

We agree with this issue, and we referred to the alternative uses of maize crop in the Introduction (lines 44-45 pag 1). We considered to not go more in depth with this argument, but it was impossible to not mention it at all.

We must not forget that maize planting is a very intensive form of crop farming and involves the high use of water.

Thank you very much for this remark. We completely agree with the reviewer’s comment, and we have added this crucial point in the Introduction (lines 159-161 and 178 - 181 pag 2) and Results and Discussion (lines 814-816 pag 7), both supported by reference 13.

Apart from the details of the modest covered data of two growing seasons and crops alternatives, the design of the research is correct. The methodology is well developed and explained. There is no objection.

Thank you for considering this experiment robustly carried out and well explained. We hope this research can be useful for other scientists (especially young ones) in their further studies.

What I saw as missing from this research is a Discussion section. My focus should be on the lack of data due to the short time of collecting results, and the alternatives in order to satisfy the same demand.

In the original manuscript we produced a jointed Results and Discussion section. Despite the sentences for discussing each parameter are still present in text, the section was renamed as Results. Thank you for this correction, we will point it out with the assigned editor.

At least, I should expect some reference to other alternative crops, water irrigation needs, and crop prices. In order to complete the research. It is a matter to compare efficiency to effectiveness.

We referred to the topics of alternative crops and crop prices through references number 3 and 15, while considering the maize water requirements we used reference 13. We did not provide any specific references for water irrigation needs because the experiment focused more on the climatic adaptability of the crop, and in this sense, we provided many references (1, 4, 5, 7, 9, 10, 11, 12, and 14, just in introduction).

Interesting study in line with the agricultural practices for adaptation to climate change and mitigating climate change.

Thank you again for the positive evaluation of this work.

Reviewer 2 Report

Interesting study in line with the agricultural practices for adaptation to climate change and mitigating climate  change.

Comments:

  • According to the section on materials and methods, sweet maize (Zea mays var. saccharata L., hybrid Centurion F1) was included in the study. There is no such information in the title of the work, which suggests that it concerns the cultivation of the subspecies Zea mays var. indurata - the specificity and purpose of sweet maize cultivation is different.
  • In the introduction to the work, reference should be made to sweet maize and the specificity of growing this subspecies. In the current version of the introduction, we have general information about the cultivation of maize. Also, the conclusion lacks information that the work concerns sweet maize.
  • Why in table 3 there are RN, DN and FN symbols and in the description in the text they are RC, DI and FI. The situation is similar in Figure 4 - different symbols than in the text.
  • In references position 14 is the abbreviation of the journal name – Eur J Agron and position 15 is the full name – European Journal of Agronomy.
  • At references position 33 – “J Agron” instead of “Agron J”.
  • Similarly at references 53: Agricultural water management, 2021, 244 - improve on: Agric Water Manag 2021, 244

Author Response

Comments:

  • According to the section on materials and methods, sweet maize (Zea mays var. saccharata L., hybrid Centurion F1) was included in the study. There is no such information in the title of the work, which suggests that it concerns the cultivation of the subspecies Zea mays var. indurata - the specificity and purpose of sweet maize cultivation is different.

Thank you for this remark, the title of the manuscript is now: Adaptive agricultural strategies for facing water deficit in sweet maize production: a case study of a semi-arid Mediterranean region.

  • In the introduction to the work, reference should be made to sweet maize and the specificity of growing this subspecies. In the current version of the introduction, we have general information about the cultivation of maize. Also, the conclusion lacks information that the work concerns sweet maize.

Thank you very much for this remark. We completely agree with the reviewer’s comment, and we have added this crucial point in the Introduction (lines 159 – 161 and 179-181 pag 2) and Results and Discussion (lines 812-814 and 814-816 pag 7). Those points are corroborated by references 5 and 13.

  • Why in table 3 there are RN, DN and FN symbols and in the description in the text they are RC, DI and FI. The situation is similar in Figure 4 - different symbols than in the text.

We did a typing error, RN, DN and FN refers to an old codification of the treatments. Now, Table 3 and Figure 4 are properly corrected. Thank you for noticing it.

  • In references position 14 is the abbreviation of the journal name – Eur J Agron and position 15 is the full name – European Journal of Agronomy.

We corrected it. Currently, both the references have the abbreviation instead of the full name.

  • At references position 33 – “J Agron” instead of “Agron J”.

Updated with the right journal abbreviation.

  • Similarly at references 53: Agricultural water management, 2021, 244 - improve on: Agric Water Manag 2021, 244

Corrected, thanks.

Thank you for your thorough and valid comments.